# Picosecond to millisecond tracking of a photocatalytic decarboxylation reaction provides direct mechanistic insights

Aditi Bhattacherjee [1,3]*, Mahima Sneha [1], Luke Lewis-Borrell [1], Omri Tau[1], Ian P. Clark[2] & Andrew J. Orr-Ewing [1]*

The photochemical decarboxylation of carboxylic acids is a versatile route to free radical intermediates for chemical synthesis. However, the sequential nature of this multi-step reaction renders the mechanism challenging to probe. Here, we employ a 100 kHz mid-infrared probe in a transient absorption spectroscopy experiment to track the decarboxylation of cyclohexanecarboxylic acid in acetonitrile-$d_3$ over picosecond to millisecond timescales using a photooxidant pair (phenanthrene and 1,4-dicyanobenzene). Selective excitation of phenanthrene at 256 nm enables a diffusion-limited photoinduced electron transfer to 1,4-dicyanobenzene. A measured time offset in the rise of the $CO_2$ byproduct reports on the lifetime (520 ± 120 ns) of a reactive carboxyl radical in solution, and spectroscopic observation of the carboxyl radical confirm its formation as a reaction intermediate. Precise clocking of the lifetimes of radicals generated in situ by an activated C-C bond fission will pave the way for improving the photocatalytic selectivity and turnover.

[1] School of Chemistry, University of Bristol, Cantocks Close, Bristol BS8 1TS, UK. [2] Central Laser Facility, Research Complex at Harwell, Science and Technology Facilities Council, Rutherford Appleton Laboratory, Harwell Oxford, Didcot, Oxfordshire OX11 0QX, UK. [3] Present address: AMOLF, Science Park 104, 1098 XG Amsterdam, The Netherlands. *email: a.bhattacherjee@amolf.nl; a.orr-ewing@bristol.ac.uk

The photochemical decarboxylation of free aliphatic carboxylic acids has recently emerged as a robust means to drive varied chemical reactions enabling the formation of complex molecular architectures and adducts[1–4]. A number of advantages—light-induced reaction offering a clean driving source, circumvention of toxic metal catalysts, use of milder reaction conditions and catalytic amounts of the photooxidants, and product formation in high yields—render it a powerful alternative to other well-established reactions such as the Barton decarboxylation[5]. The catalytic photochemical decarboxylation makes use of single electron transfer between an electron donor (a photoexcited species such as an arene) and electron acceptor (e.g. an organic nitrile) pair to form ionic intermediates. The arene cation is thought to be the active oxidant of the carboxylic acid, although a similar role played by the photoexcited electron acceptor is not completely excluded[6]. The radical formed from the decarboxylation is a useful precursor for chemical products featuring new terminal carbon–hydrogen bonds and self/cross-coupling carbon–carbon bonds[7].

Figure 1a shows the reactants used in this study for spectroscopically tracking the stepwise reaction mechanism and kinetics of a photocatalytic decarboxylation reaction. The co-oxidant pair is phenanthrene (arene, abbreviated PHEN) and 1,4-dicyanobenzene (electron-acceptor, abbreviated DCB) in acetonitrile-$d_3$ (solvent, abbreviated ACN-$d_3$). The deuterated solvent is chosen as the reaction medium in order to enable greater infrared (probe) transmission in the region of the ring distortion modes. Cyclohexanecarboxylic acid (abbreviated CHCA) is used in an alkaline medium (NaOH solution) as the decarboxylation of the carboxylate anion is reported to be more efficient in comparison to the carboxylic acid[8]. Synthetic efforts that make use of this

strategy for the generation of new C–C/C–H bonds report the formation of the decarboxylated product in excellent yields (> 97%) and near-quantitative (> 95%) recovery of PHEN and DCB after performing the reaction in an inert atmosphere for over 6 h[8]. However, with the current research focus on increasing the reaction substrate scope and selectivity, a quantitative mechanistic understanding of the elementary steps occurring in the reaction mixture is lacking.

A plausible reaction pathway (Fig. 1b), proposed by Yoshimi et al.[8], involves photoexcitation of PHEN to generate PHEN* as the driving step (Step 1). In organic synthesis, this is typically achieved using a high-power mercury arc lamp together with a pyrex filter, restricting the excitation wavelengths to longer than ≈280–300 nm. An excited-state, single electron transfer (SET, Step 2) from PHEN* to DCB leads to the formation of Phen$^+$-DCB$^-$, likely born as a geminate ion pair and transforming to a solvent-separated ion pair before being fully solvated to produce free ions in solution[9] (note that 'ions' are often referred to as 'radical ions' in the literature; however, we adhere to the use of 'ions' in our discussion throughout). As supporting evidence, the presence of water in the reaction medium is reported to lower the efficacy of back electron transfer by facilitating the formation of free solvated ions[10]. A second oxidative SET (Step 3) between a carboxylate anion (RCOO$^-$) and the arene cation (PHEN$^+$) generates a reactive carboxyl radical (RCOO$^\bullet$) in solution. This radical precursor then undergoes a unimolecular dissociation (Step 4) to form R$^\bullet$ by releasing $CO_2$ as a byproduct. A potential role of the electron acceptor (DCB) as the active photooxidant of the carboxylate ion is invoked[6], but largely unclear.

Here, we perform a spectroscopic tracking of the sequential SET steps leading up to the evolution of $CO_2$ using temporally-delayed mid-infrared pulses with a dynamic range of sub-picosecond to several hundred microseconds in a time-resolved multiple probe experiment (≈200 fs instrument response)[11–15]. A narrowband (< 100 cm$^{-1}$ bandwidth), ultrashort laser pulse at 256 nm is employed to excite PHEN selectively in the reaction mixture. The progress of the decarboxylation reaction is probed in the combined regions of the nitrile stretch (≈2100 cm$^{-1}$), carboxyl stretch (≈1700 cm$^{-1}$), C–C stretch/C–H bend (≈1500 cm$^{-1}$), and $CO_2$ antisymmetric stretch ($\nu_3$, ≈2340 cm$^{-1}$). Time-resolved electronic spectroscopy using a white-light probe (350–700 nm) is additionally used to infer the ultrafast relaxation and electron transfer occurring within the first few hundred picoseconds. A comparison of the fits for the measured decay of the cationic intermediate and the rise of $CO_2$ using a three-state sequential kinetic model is used to characterize the unimolecular decomposition of the carboxyl radical species by C–C bond fission. The kinetic analysis is also supported by a direct spectroscopic observation of the carboxyl radical intermediate. These results constitute a determination of the lifetime of a dissociating carboxyl radical in a photocatalytic reaction in situ.

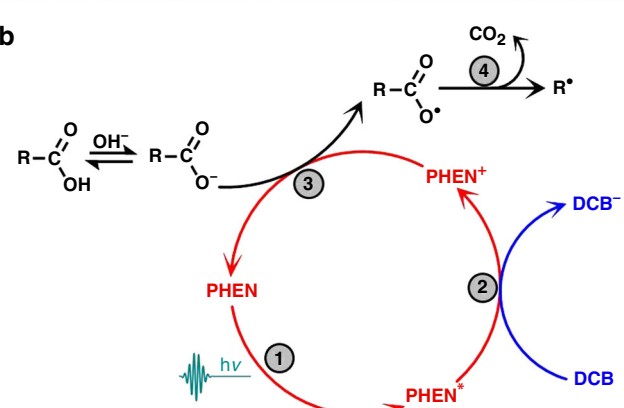

**Fig. 1** A representative photocatalytic decarboxylation reaction. **a** Chemical structures of the molecules involved in the photochemical decarboxylation reaction—Phenanthrene (PHEN), 1,4-Dicyanobenzene (DCB) and Cyclohexanecarboxylic acid (CHCA). **b** Schematic diagram of a light-induced sequential decarboxylation reaction. Steps 1 through 4 denote photoexcitation, bimolecular single electron transfer from PHEN* to DCB, oxidative single electron transfer from carboxylate anion, and a C–C bond fission that produces a free radical and $CO_2$ byproduct, respectively. An ultrashort pulse (256 nm, h$\nu$ = 4.8 eV) is used to initiate the reaction in the present study

## Results

**Photoexcitation and single electron transfer.** Figure 2 shows the time-resolved infrared (TRIR) spectra (panels a–d) and kinetics (panels e–f) in the regions of the in-plane ring distortion modes plus C–H bends (1450–1600 cm$^{-1}$, Fig. 2a, c) and antisymmetric nitrile stretch (2060–2140 cm$^{-1}$, Fig. 2b, d), measured after 256 nm photoexcitation of a solution of PHEN (8 mM) and DCB (30 mM) in ACN-$d_3$. These spectra show changes in the optical density of the sample triggered by the photoinduced reaction, against a logarithmic timescale in the top panel and over distinct time-windows in the bottom panel. The experiment is also carried out with an initially equimolar ratio of DCB and PHEN (10 mM each), with a stepwise-increased concentration of DCB

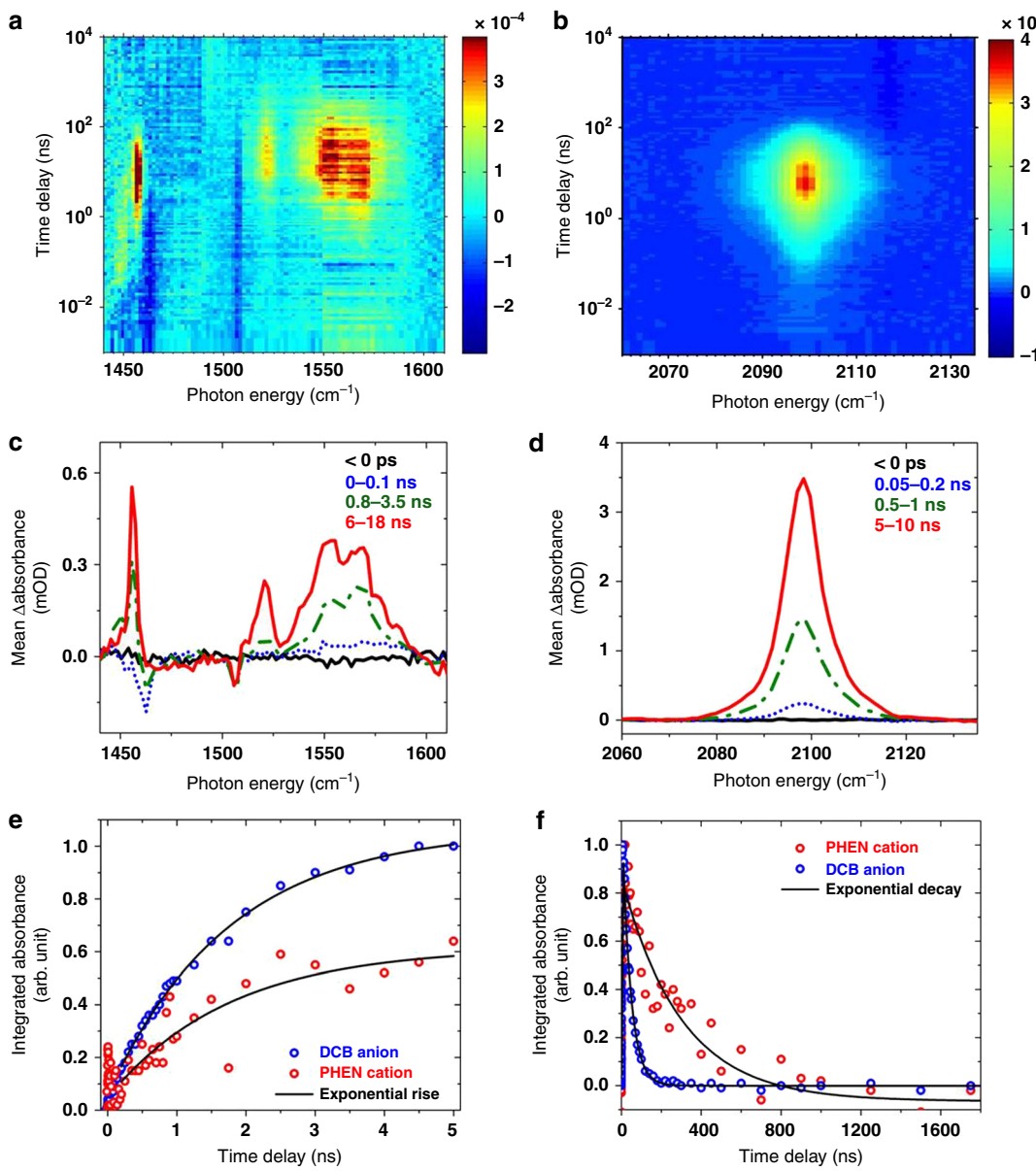

**Fig. 2** Single electron transfer from PHEN* to DCB. **a, b** 2D false color maps of the time-resolved infrared (TRIR, or transient infrared) spectra of a mixture of PHEN (~8 mM) and DCB (~30 mM) in acetonitrile-$d_3$ which is photoexcited at 256 nm. Note the logarithmic scale of the time axis. Amplitude units are in differential optical density (ΔOD), as annotated by the color bar on the right. **c, d** TRIR spectra in the top panels are shown as evolving over representative time windows (solid black line measured at negative time delays is a reference for the baseline; dotted blue line is for 0–100 ps in **c** and 50–200 ps in **d**; dash-dotted green line is for 0.8–3.5 ns in **c** and 0.5–1 ns in **d**; solid red line denotes 6–18 ns in **c** and 5–10 ns in **d**). The decay of the peaks on longer timescales, evident in the top panel, is not shown here for clarity. **e** Rise in the integrated absorbance of the peaks between 1520 and 1580 cm$^{-1}$ (hollow red circles, PHEN$^+$) and 2098 cm$^{-1}$ (hollow blue circles, DCB$^-$) is fit to a global single-exponential time constant of 1.7 ± 0.1 ns. Note that the overlapping peaks due to PHEN* have weaker absorption and do not dominate the kinetics beyond 100 ps. **f** Decay in the integrated absorbance of the same peaks shows different exponential decay time constants for DCB$^-$ (42 ± 1 ns) and PHEN$^+$ (300 ± 30 ns)

using a 100 mM stock solution in order to measure the pseudo first-order electron transfer rate coefficient. The dataset shown in Fig. 2 represents the highest concentration of DCB at which the reaction is probed and is chosen for the best signal-to-noise ratio.

A depletion of the ground-state absorption bands of PHEN (1462 and 1506 cm$^{-1}$) and a rise of excited state (PHEN*) absorption features (unresolved peaks between 1550–1580 cm$^{-1}$) are discernible immediately upon photoexcitation (Step 1 in the introduced reaction scheme). These early-time (< 1 ps), weak (< 200 μOD) features are competitive with the sub-picosecond instrument response of the apparatus and are additionally probed

using time-resolved electronic spectroscopy (Supplementary Figs. 1 and 2, Supplementary Note 1). Nonetheless, the absence of any peak in the nitrile stretching region due to DCB* (expected position 2135 cm$^{-1}$, see Supplementary Fig. 3 for photoexcitation at 280 nm) presents unambiguous evidence for the selective excitation of PHEN in the reaction mixture at 256 nm despite significant overlap of the UV-VIS absorption spectra of the two compounds (Supplementary Fig. 4, Supplementary Note 2). We note that at wavelengths longer than 280 nm, as often used in organic synthesis experiments[16,17], photoexcitation of DCB is likely to occur as the absorption cross-section is competitive with PHEN. Narrowband excitation at 256 nm enables us to perform

the reaction in a clean and controlled manner to pin down the reaction mechanism.

Post 100 ps, weak to moderately weak absorption bands (≤ 0.6 mOD and ≥ 1 mOD, respectively) can be identified in both the spectral regions (Fig. 2a, b) which are assigned to the infrared modes of the ions (PHEN$^+$; 1520, 1540–1585 cm$^{-1}$ and DCB$^-$; 1456, 2098 cm$^{-1}$) formed due to a single electron transfer from PHEN* to DCB (Step 2). The nitrile stretch undergoes a shift to lower energies by >130 cm$^{-1}$ in the anion of DCB compared to that of the ground-state. The ring-distortion modes are not significantly affected, however. The assignment, consistent with previous observations of the nitrile stretch of the DCB anion[18], vibrational frequency calculations (Supplementary Fig. 5), and control experiments with anthracene (Supplementary Fig. 6), reflects the formation of an ion pair in solution.

**Rise and decay of the photoinduced ion pair**. The kinetics of the single electron transfer (Step 2) are accurately reflected through a global fit of the rise in absorptions of the antisymmetric nitrile stretching frequency of the DCB anion (2098 cm$^{-1}$) and the in-plane ring distortion/C–H bend modes of the PHEN cation (≈1540 cm$^{-1}$), shown in Fig. 2e. This combined analysis gives an exponential time-constant of 1.7 ± 0.1 ns for the formation of PHEN$^+$ and DCB$^-$ under these reaction conditions. The sharp rise and fall (≤ 200 ps) evident in the red trace is attributed to PHEN* and excluded from the fit to obtain reliable photoinduced electron transfer rates. Note that at these millimolar-level concentrations of DCB and PHEN in a non-viscous solvent, the generalized Smoluchowski theory is well approximated by the use of exponential kinetics[19,20]. A study of the photoinduced electron transfer rates with varying concentrations of DCB (Supplementary Fig. 7) affords the estimation of a pseudo first-order SET rate coefficient of 1.9(3) × 10$^{10}$ M$^{-1}$s$^{-1}$. This value is commensurate with an expected diffusion-controlled rate constant of 1.9 × 10$^{10}$ M$^{-1}$s$^{-1}$ in ACN at 25 °C [21].

The fate of the nascent ion pair can be observed through the decay traces shown in Fig. 2f. It is clear that even without the addition of the carboxylic acid, the ions decay in the solution over tens to hundreds of nanoseconds. Furthermore, there is an evident disparity in the decay time-constants, indicating different loss mechanisms are in operation for the cation and the anion: DCB$^-$ decays with a time-constant of 40 ± 2 ns, whereas PHEN$^+$ decays with a time-constant of 340 ± 70 ns (error bars denote the standard deviation over seven measurements). Thus, the decay of the ion pair cannot be solely due to charge recombination. We suggest the main decay pathway is collisional quenching with dissolved oxygen, i.e., (i) $DCB^- + O_2 \rightarrow DCB + O_2^-$ and (ii) $PHEN^+ + O_2^- \rightarrow PHEN + O_2$, found to be persistent in our samples despite purging with nitrogen. In fact, oxidative quenching of related nitroaromatic anions by molecular oxygen to yield a superoxide ion is notoriously referred to as 'futile reduction' by nitroreductases[22]. The PHEN cation might be reduced by the superoxide ion to recover the ground-state molecule, which could explain the longer lifetime of PHEN$^+$ compared to DCB$^-$. Other minor channels for the loss of PHEN$^+$ may be photosubstitution, polycyclic aromatic hydrocarbon formation[23], unimolecular dissociation, or reaction with excess electrons in solution. Carrying out the reaction in an inert argon atmosphere, as reported in organic synthesis studies[8], might be crucial for higher decarboxylation reactive yields because of extended PHEN$^+$ lifetimes but does not affect the conclusions of the present study (see Supplementary Note 3).

**Photocatalytic decarboxylation**. Figure 3 shows the TRIR spectra measured in the region of the antisymmetric stretch of carbon

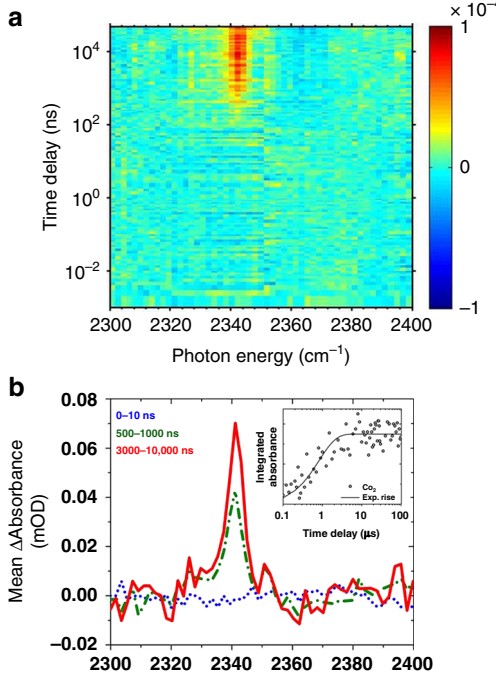

**Fig. 3** Time-resolved photocatalytic decarboxylation. **a** False color map of the TRIR spectrum of the full reaction mixture—PHEN (4 mM): DCB: CHCA: NaOH = 1:4:6:6—photoexcited at 256 nm and measured in the region of the antisymmetric stretch of the reaction byproduct (CO$_2$). The time axis is plotted on a logarithmic scale for clarity. Amplitude units, highlighted by the color map, are in ΔOD. **b** Transient absorption spectra averaged over representative time-windows (0–10 ns in dotted blue, 500–1000 ns in dash-dotted green, 3–10 µs in solid red). The inset shows a kinetic trace of the integrated absorbance of this peak on a logarithmic time-axis, and fitted to a single-exponential function

dioxide, for a 256-nm-photoexcited solution of PHEN (4 mM), DCB (16 mM), CHCA (24 mM), and NaOH (24 mM). We note that CHCA has very weak absorption in the ultraviolet, and most notably at wavelengths below 240 nm (Supplementary Fig. 4). Therefore, any reaction product probed arises from the decarboxylation cycle (Fig. 1b) without interference from the nascent photochemistry of the acid[24,25]. CHCA is expected to be only weakly dissociated in solution (pK$_a$ = 7.6)[26]; an equimolar solution of sodium hydroxide favors an equilibrium lying towards the carboxylate anion. At the same time, the experiment avoids a higher concentration of sodium hydroxide to prevent the precipitation of the sodium carboxylate salt due to its low solubility.

The false color map shown in Fig. 3 reveals the formation of carbon dioxide as the reaction byproduct as early as 100 ns after photoexcitation of PHEN. The observed peak center of the antisymmetric stretch (2341 cm$^{-1}$, ≈70 µOD) is remarkably similar to that found in aqueous solutions of CO$_2$ and not much different from the isolated molecule itself[27]. This invariance suggests minimal interaction with the solvent. The observation of only one component of the band suggests a hindered rotation of the molecule in the solvent cage[28]. A kinetic trace of the integrated absorbance of the peak, fitted to an exponential rise function, indicates a sub-microsecond rise of the reaction byproduct. This measurement corresponds to a real-time observation of the decarboxylation reaction which is the fourth and final step introduced earlier in the reaction scheme (Fig. 1b).

A point of contention in the literature is whether DCB is an active photooxidant for the carboxylate anion[6]. A control

experiment (Supplementary Fig. 6) replacing PHEN with anthracene at similar concentrations shows no sign of $CO_2$ evolution within the signal-to-noise ratio of the experiment. This is expected because the secondary electron transfer from a carboxylate ion (e.g. hexanoate ion) to PHEN$^+$ is a spontaneous process ($\Delta G = -0.34$ V in ACN) but energetically uphill for the anthracene cation ($\Delta G = +0.07$ V in ACN)[8,21]. This, together with the absence of DCB* bands in the TRIR spectra at 256-nm photoexcitation, presents strong evidence that under these reaction conditions PHEN$^+$ takes part in the third step in an oxidative collision via a SET. We note that in organic chemistry reactions employing broadband excitation sources where photo-excitation of DCB→DCB* can occur, decarboxylation of the cation of the carboxylic acid formed via a SET from the neutral carboxylic acid to the excited state of the electron acceptor (DCB*) is possible[29]. Additionally, decarboxylation of a carboxylate anion by electron transfer to a photoexcited DCB sensitizer is also reported to occur[30]. We carried out a control experiment with DCB, photoexcited at 280 nm (in the absence of PHEN), in ACN-d$_3$ containing CHCA/NaOH (24 mM). The experiment showed the formation of DCB* due to photoexcitation but no evidence for the growth of DCB$^-$ due to a SET from carboxylate anion to DCB*.

**Kinetic model and radical lifetime.** The measured lifetime of the PHEN$^+$ cation and the observed growth of $CO_2$ can be used to estimate the timescales over which the transient carboxyl radical forms and undergoes an activated C–C bond fission. We find that upon addition of an alkaline solution of CHCA (10–25 mM), the lifetime of PHEN$^+$ is marginally reduced ($230 \pm 70$ ns, over thirteen measurements, versus $340 \pm 70$ ns in a solution containing only the photooxidant system, i.e. DCB + PHEN). This change could, in part, be due to reductive quenching of PHEN$^+$ with excess hydroxyl ions. Nonetheless, a SET between the carboxylate ion and the PHEN$^+$ cation (step 3) must necessarily occur within the solution-phase lifetime of the PHEN$^+$ cation for the reaction to proceed through to the final decarboxylation step (step 4). The measured time-offset between the decay of PHEN$^+$ (step 3) and the rise of $CO_2$ (step 4) thus evidently reports on the how long the cyclohexane carboxyl radical persists in solution before undergoing a unimolecular decay (Fig. 4). Application of a three-state sequential reaction mechanism[31] (see figure caption and Supplementary Methods for details) shows that the population of the carboxyl radical peaks at $320 \pm 60$ ns with a lifetime of $520 \pm 120$ ns (error bars denote standard deviation over nine measurements).

An estimated radical lifetime of several hundred nanoseconds in solution indicates the presence of an energy barrier along the reaction coordinate. Thus, the decarboxylation reaction is activation-controlled. Typical lifetimes of carboxyl radicals in solution, made accessible by flash-photolysis[32], pulsed radiolysis[33], and time-resolved electronic paramagnetic resonance techniques[34], are reported to vary over nanosecond to microsecond timescales. Ab initio calculations show that the reaction involves passage over a transition state and is exothermic[35–40]. The reaction rates are primarily governed by resonance stabilization of the radical (e.g. for aryl groups), free energy of the transition state, single or double-bond character adjacent to the dissociating C–C bond, and influence of low-lying excited states on the ground-state potential energy surface. We compute an energy barrier of 7 kJ mol$^{-1}$ for the in vacuo conversion of cyclohexanecarboxyl radical to cyclohexyl radical (Supplementary Fig. 8). Although the decarboxylation reaction is expected to be exothermic (by 65 kJ mol$^{-1}$), there is no evidence for a large excess of vibrational energy deposited in the evolving $CO_2$ molecule. This is because the

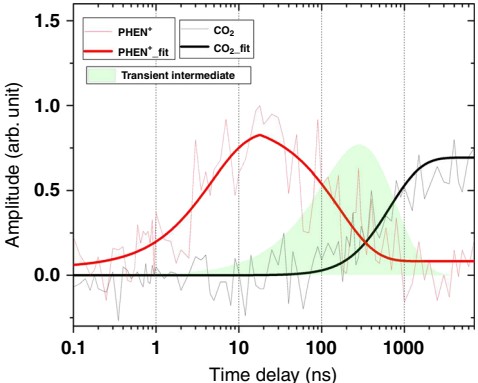

**Fig. 4** Photodynamics in a decarboxylation reaction. Decarboxylation kinetics fitted according to a three-state sequential reaction pathway (A → B → C). A, B, and C denote RCOO$^-$, RCOO$^\bullet$, and R$^\bullet$ (or equivalently, $CO_2$), respectively, in the introduced reaction scheme (Fig. 1b). The model assumes that the decay of the CHCA anion is limited by the total (reactive + non-reactive) decay of PHEN$^+$ (red trace), as required by a bimolecular reactive collision between the two partners (step 3, Fig. 1b). The time constant for the decay of the PHEN$^+$ cation (equivalently, decay of the CHCA anion) is $170 \pm 40$ ns (red trace), and that for the rise of the $CO_2$ byproduct is $555 \pm 85$ ns (black trace) in this measurement. Error bars denote one standard error in the fitting procedure. The kinetic model, applied to this dataset, predicts that the population of the transient cyclohexane carboxyl radical (filled-green trace) peaks at ≈280 ns

dissociation occurs adiabatically and (picosecond) vibrational cooling by the solvent bath efficiently thermalizes the dissociating fragments. Thus, a photocatalytic decarboxylation is expected to be a much softer technique for manipulating radical chemistry, exercising a better degree of control over the final photoproducts, in comparison to the direct photoexcitation of the carboxylic acid or its charge-transfer complexes.

A low infrared probe transmission in the region of the asymmetric carboxyl stretch of the radical intermediate (due to the nascent IR absorption bands of the carboxylic acid/carboxylate anion) may impede its direct observation in a pump-probe experiment. Furthermore, the use of low UV pump power (80 nJ) to prevent multiphoton ionization of PHEN in the measurements outlined above may also have rendered the radical intermediate inaccessible to experimental observations. With the use of ≈tenfold higher levels of pump power, we could observe a band at 1735 cm$^{-1}$ whose wavenumber value and time-dependence support its assignment to the carboxyl stretch of the cyclohexanecarboxyl radical. We find that the rise and decay of this band is fully consistent within the framework of the proposed three-state sequential kinetic model, that is, the radical indeed appears as an intermediate (Supplementary Fig. 9, Supplementary Note 4) between the decay of PHEN$^+$ and the rise of the reaction byproduct, $CO_2$. This consistency illustrates the general applicability of a kinetic model to extract the lifetimes of reactive intermediates reliably, even if not directly observed in an experiment. Further, we propose that probing the $CO_2$ byproduct, whose IR absorption frequency appears in a clean region (that is, without overlapping absorption by carboxyl/ring distortion modes of the reagents), makes it suitable for tracking the photocatalytic decarboxylation of other aliphatic and aromatic carboxylic acids in situ.

## Discussion
The multi-step reaction mechanism of a photoredox-catalyzed oxidative decarboxylation is tracked spectroscopically in real time. Selective excitation of a ground-state PHEN molecule in the reaction mixture enables the study of the reaction in situ in a clean

fashion, without interference from the photochemical pathways of the other reactants. The elementary steps outlined in the photoredox cycle, from photoexcitation to bimolecular electron transfer and decarboxylation, are directly observed and the kinetics of the multiple steps are characterized over eight orders of magnitude in time. Control experiments using anthracene as the photooxidant or DCB* in the absence of PHEN clearly establish that the arene cation is reponsible for the oxidative decarboxylation under these experimental conditions. The observed timescales for the decay of the photooxidant PHEN$^+$ and growth of $CO_2$ inform our knowledge of the lifetime (hundreds of nanoseconds) of a transient carboxyl radical in solution. The results afford a combined kinetic and mechanistic understanding of a four-step photoredox chemical reaction over picosecond to millisecond dynamic time range in a single time-resolved experiment.

Our work demonstrates the value of transient absorption with high repetition-rate spectroscopic probes to resolve multi-step photocatalytic reaction mechanisms[41]. A precise knowledge of the mechanistic pathway and lifetimes of the transient radicals or intermediates, afforded by such studies, will lead to further innovation in the design of photoredox-catalyzed, self-, and crosscoupling reactions. The methodology outlined here as a proof-of-principle experiment can be readily extended to other photoredox chemical reactions that involve multiple steps or intermediates, such as organic polymerization reactions triggered by the photoinduced generation of free radicals. Our experimental scheme additionally reveals that organic synthetic methods, which traditionally use high power, broadband arc lamps for high product yields may further benefit from low power, narrowband photoexcitation for implementing selectivity in photoredox reactions.

## Methods

**Infrared spectroscopy measurements**. Time-resolved infrared spectroscopy is performed using ultraviolet (pump) and mid-infrared (probe) optical parametric amplifiers driven by a 100-kHz Yb:KGW ultrafast amplifier. Details of the experimental apparatus are oulined elsewhere[11]. Briefly, 256 nm pump-pulses (80 nJ per pulse, 1 kHz) are focused to a 150-μm diameter spot size inside a continually flowing solution of 100 μm path length. Two separately-tunable, broadband midinfrared probe pulses (≤ 0.05 μJ per pulse, 200 cm$^{-1}$ bandwidth, 100 kHz) are focused to a spot-size of 50–75 μm in the interaction region of the pump beam with the sample. An advantage of this setup is that it makes use of two IR probe beams that can be used to monitor the reaction in two different spectral regions simultaneously. The residual pump is blocked and the probes are dispersed on two 128-element MCT (mercury cadmium telluride) detectors. The experiment uses 100 probe pulses for every pump pulse; the last probe pulse in this sequence provides the 'pump-off' reference measurement in order to determine the differential absorbance of the probe due to the light-induced reaction. This procedure furnishes transient absorption spectra at multiple, pre-defined time delays. The pump-probe time delay is scanned over 8 ns using an optical delay line, between 8 ns and 9 μs using electronic delays, and makes use of the repetition-rate of the laser to probe the reaction every 10 μs, out to 1 ms.

## Data availability

Data are available at the University of Bristol data repository, data.bris, at https://doi.org/10.5523/bris.10.5523/bris.1q3gfn8eb06hh22sjzql2cu1qg. All reported data are available from the corresponding authors upon reasonable request.

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

## Acknowledgements

This research is funded by EPSRC Grant No. EP/R012695/1 and a previously completed ERC Advanced Grant No. CAPRI 290966. M.S., L.L.B. and A.J.O.E. gratefully acknowledge a Marie-Curie Fellowship (MARCUS 793799) and the Bristol Chemical Synthesis Centre for Doctoral Training (EPSRC EP/L015366/1). We are grateful to the Science and Technology Facilities Council for access to the LIFEtime and ULTRA facilities.

## Author contributions

The research idea was conceived by A.J.O.E. A.B. designed and planned the experiments. A.B. conducted the experiments—at the University of Bristol with M.S. and at the LIFEtime facility with M.S., L.L.B. and I.P.C. O.T. initiated the work as part of an undergraduate research project. A.B., M.S., O.T. and A.J.O.E. analyzed all or parts of the experimental and computational data. A.B. wrote the paper with input from A.J.O.E. All authors agree with the final version of the paper.

## Competing interests

The authors declare no competing interests.
