## [Peer Review File · Nature Communications]

Reviewers' comments:

Reviewer #1 (Remarks to the Author):

The manuscript by Orr-Ewing and co-authors documents a time-resolved, spectroscopic study of a photo-catalytic reaction over many orders of magnitude of time. The photo-catalytic reaction is decarboxylation of CHCA, thought to involve a reactive carboxyl radical important to many organic reactions. The results are new and of interest to the community, and therefore overall, I recommend publication. However, I have one major issue, involving a central claim of the manuscript, and a couple of important technical issues that should be addressed in order to warrant publication.

The major issue surrounds the directness of the claim that the lifetime of the carboxyl radical is determined by the time-offset between PHEN⁺ decay and CO₂ rise. While this can certainly be justified in a kinetic model, the purpose of the manuscript is to directly (spectroscopic-ally) report on reaction intermediates, along with reactants and products. The authors should at the very least: (1) accommodate the language ("a measured radical lifetime", "the radical must necessarily occur"-- maybe, but if it hasn't been observed, and this is in part the point of the measurements, it seems strong) (2) explain why it is difficult to observe this radical, if it indeed occurs over the time-period specified, and (3) The claim is also made more difficult by the PHEN⁺/DCB- decay in the absence of CHCA, attributed to O₂ quenching. How this quenching reaction interferes with extracting the lifetime of the carboxyl radical should be more extensively discussed, and possibly quantified (e.g. error bars on the lifetime?).

The following two technical issues should also be addressed:

(1) Why is the rise and decay of the PHEN⁺ reported with an integrated absorbance that includes PHEN* (1550-1580 cm⁻¹)? It seems a better and cleaner peak for PHEN⁺ is 1520 cm⁻¹.

(2) It is not clear how the kinetic model for the radical lifetime includes the O₂ quenching or decay pathway in the absence of CHCA. I believe it should be included if that is a competing pathway. Since this is an additional limitation to directly measuring the lifetime of the radical, it should be shown/discussed in the main manuscript.

Reviewer #2 (Remarks to the Author):

This manuscript presents an interesting time-resolved spectroscopy investigation employing a 100 kHz mid-infrared probe in a transient absorption spectroscopy experiment to directly observe the decarboxylation of cyclohexanecarboxylic acid (CHCA) in acetonitrile-d₃ from the picosecond to the millisecond timescales by monitoring a photooxidant pair (phenanthrene, PHEN and 1,4-dicyanobenzene, DCB). The selective excitation of PHEN in the reaction mixture at 256 nm allows a diffusion-limited photoinduced electron transfer to DCB and an observed time offset in the rise of the CO₂ byproduct with respect to the decay of the oxidizing arene cation directly measures the lifetime (520 ± 120 ns) of a reactive carboxyl radical in this photocatalytic decarboxylation reaction. This ability to perform accurate spectroscopic lifetime measurements of the radicals produced in situ by an activated C-C bond fission can be utilized to improve photocatalytic selectivity and turnover in related reactions. The experiments and their data analysis and interpretation all have

appeared to have been carefully performed and considered. I recommend publication after minor revision to address the comments/suggestions below:

1. It would be nice to include some related computational work to support the assignments of the species observed in the time-resolved experiments and to also explore the reaction pathways and barriers proposed for the interpretation of the experimental data. I would encourage the authors to do this to make the research both stronger and more interesting to a broader readership so as to make it more suitable for Nature Communications.

Reviewer #3 (Remarks to the Author):

Time-resolved infrared spectroscopy is applied to study the photochemical decarboxylation of a particular carboxylic acid (CHCA) in acetonitrile using a photooxidant organic donor - acceptor pair, phenanthrene (PHEN) and 1,4-dicyanobenzene (DCB). This reaction is studied over about 12 orders of magnitude in time, from picoseconds to milliseconds.

The general selling point is that by understanding the reaction mechanisms may be of use in improving molecular design of supramolecular structures. The procedure is argued to occur under milder reaction conditions, using a clean phototrigger, and the use of toxic metal catalysts is avoided. This may be so, however, after the introduction and a lengthy detailed description of the spectroscopic results, this general context is not being mentioned anymore. What does one learn from the results obtained for future applications?

Despite the extensive description of the experimental results and the interpretations of those, I sense that many issues remain unmentioned unanswered:

1-what are the driving forces for the two charge transfer reactions

$\text{PHEN}^* + \text{DCB} \rightarrow \text{PHEN}^{\cdot+} + \text{DCB}^{\cdot-}$

$\text{PHEN}^{\cdot+} + \text{CHCA}^- \rightarrow \text{PHEN} + \text{CHCA}$

would the observed rates makes sense when comparing previously reported results of photoinduced electron transfer reactions of organic donor-acceptor pairs in acetonitrile, as extensively documented by the Vauthey group

2- Fig. 1 suggests unit quantum yields for the sequential charge transfer steps. I miss a dedicated discussion of and a quantitative assessment of quantum yields the two charge transfer reactions.

The authors need to come with convincing arguments why a back-electron transfer reaction

$\text{PHEN}^{\cdot+} + \text{DCB}^{\cdot-} \rightarrow \text{PHEN} + \text{DCB}$

does not form a serious contender cutting short the decarboxylation process.

3- CHCA is prepared in the anionic form CHCA^- using equimolar amounts of NaOH. May I assume this is anhydrous NaOH? Even then, How much exactly of CHCA^- is in the anionic form? And because of that one expects to have the same amount of H_2O in solution as well. What is the role of H_2O in the decarboxylation process?.. Is it possible to investigate possible H/D isotope effects (by using NaOD/ deuterated CHCA^-) to explore the role of water hydrogen bonded to a possible $\text{PHEN}^{\cdot+} / \text{CHCA}^-$ reactive complex.

Reviewers' comments:

Reviewer #1 (Remarks to the Author):

The manuscript by Orr-Ewing and co-authors documents a time-resolved, spectroscopic study of a photo-catalytic reaction over many orders of magnitude of time. The photocatalytic reaction is decarboxylation of CHCA, thought to involve a reactive carboxyl radical important to many organic reactions. The results are new and of interest to the community, and therefore overall, I recommend publication. However, I have one major issue, involving a central claim of the manuscript, and a couple of important technical issues that should be addressed in order to warrant publication.

Comment: The major issue surrounds the directness of the claim that the lifetime of the carboxyl radical is determined by the time-offset between PHEN⁺ decay and CO₂ rise. While this can certainly be justified in a kinetic model, the purpose of the manuscript is to directly (spectroscopic-ally) report on reaction intermediates, along with reactants and products. The authors should at the very least: (1) accommodate the language ("a measured radical lifetime", "the radical must necessarily occur"--maybe, but if it hasn't been observed, and this is in part the point of the measurements, it seems strong) (2) explain why it is difficult to observe this radical, if it indeed occurs over the time-period specified, and (3) The claim is also made more difficult by the PHEN⁺/DCB- decay in the absence of CHCA, attributed to O₂ quenching. How this quenching reaction interferes with extracting the lifetime of the carboxyl radical should be more extensively discussed, and possibly quantified (e.g. error bars on the lifetime?).

Response: The referee's main critique inspired us to launch renewed efforts to observe the carboxyl radical in the infrared transient absorption experiment. The main experimental difficulty with the detection of the radical earlier (#2 raised by the referee) was the low infrared probe transmission in the region of its carboxyl band. This is because the carboxyl stretches of the cyclohexanecarboxylate anion and cyclohexanecarboxylic acid (reagents employed in the decarboxylation reaction) lie in the vicinity of cyclohexanecarboxyl radical (the reaction intermediate). The low UV pump power (80 nJ at 256 nm) used in the measurements (to ensure single-photon excitation of PHEN) also made it difficult to observe the radical.

We have now carried out additional experiments using modified conditions (higher pump powers of 800 nJ at 256 nm and using a closed-flow system for reduced oxygen contamination) and successfully observed the radical carboxyl band at 1735 cm⁻¹. We find that the rise and decay of this band is fully consistent within the framework of the proposed three-state sequential reaction kinetic model, that is, the radical indeed appears as an intermediate between the decay and rise of PHEN⁺ and CO₂, respectively. This new experimental result is now incorporated in Figure S9. In new lines 246-262, we write *"A low infrared probe transmission in the region of the asymmetric carboxyl stretch of the radical intermediate (due to the nascent IR absorption bands of the carboxylic acid / carboxylate anion) may impede its direct observation in a pump-probe experiment. Furthermore, the use of low UV pump power (80 nJ) to prevent multiphoton ionization of PHEN in the measurements outlined above may also have rendered the radical intermediate inaccessible to experimental observation. With the use of ~tenfold higher levels of pump power, we could observe a band at 1735 cm⁻¹ whose wavenumber value and time-dependence support its assignment to the carboxyl stretch of the cyclohexanecarboxyl radical. We find that the rise and decay of this band is fully consistent within the framework of the proposed three-state sequential kinetic model, that is, the radical indeed appears as an intermediate (Figure S9) between the decay of PHEN⁺ and the rise of the reaction byproduct, CO₂. This consistency illustrates the general applicability of a kinetic model to extract the lifetimes of reactive intermediates reliably, even if not directly observed in an experiment. Further, we propose that probing the CO₂ byproduct, whose IR absorption frequency appears in a clean region (that is, without overlapping absorption by carboxyl / ring distortion modes of the reagents), makes it suitable for tracking the photocatalytic decarboxylation of other aliphatic and aromatic carboxylic acids in situ."*

(1) The language in the first instance pointed out by the referee is changed to “*an estimated radical lifetime*” as our model indeed provides an estimate and not a direct measurement. However, in the second instance we have retained the language “the radical must necessarily occur within the solution-phase lifetime of the PHEN⁺ cation for the reaction to proceed through to the final decarboxylation step” as we have control experiments on anthracene to prove this. Furthermore, we now have a direct experimental signature of the radical.

(2) As pointed out in the previous paragraph, the radical was difficult to observe because it was obscured by the nascent IR absorption of the asymmetric stretch of the carboxylate anion and the pump power used was also low. We were able to observe it at higher powers and a new Figure S9 has been added to show the carboxyl band of the radical in the spectrum. We would like to emphasize that CHCA is only a representative acid chosen for this particular photoredox cycle. In new lines 259-262 we propose that probing the CO₂ byproduct, whose IR absorption frequency appears with strong oscillator strength and in a clean region (that is, without overlapping ring distortion modes of the photooxidant or carboxyl absorption of the reagents), is more generally applicable to other aliphatic and aromatic carboxylic acids.

(3) We present evidence that PHEN⁺ is actually quenched by superoxide (O₂⁻) ion (which is formed by quenching of DCB⁻ by O₂). This quenching reaction and other loss mechanisms of PHEN⁺ shorten its lifetime. Thus, the decarboxylation reaction is controlled by the overall rate of decay of PHEN⁺ (which includes all loss mechanisms of PHEN⁺ including the main decarboxylation reaction and side reactions such as quenching due to superoxide anion). We have now included a note in the SI to explain how the quenching pathway influences the kinetic model, as follows:

“The possible loss mechanisms for PHEN⁺, reactive and non-reactive (with respect to the decarboxylation reaction), include:

- (1) PHEN⁺ + O₂⁻ → PHEN + O₂ [k₍₁₎; Non-Reactive]
- (2) PHEN⁺ + DCB⁻ → PHEN + DCB [k₍₂₎; Non-Reactive]
- (3) PHEN⁺ + OH⁻ → PHEN + ·OH [k₍₃₎; Non-Reactive]
- (4) PHEN⁺ + RCOO⁻ → PHEN + RCOO· [k₍₄₎; Reactive]

It must be noted that reactions (3) and (4) occur exclusively in the presence of CHCA and NaOH in solution. The serial numbering of these reactions (1)-(4) should not be confused with steps 1-4 introduced in Figure 1 of the main paper. To distinguish between these representations, we use parentheses (1)-(4) in our discussion here pertaining to the possible loss mechanisms for PHEN⁺.

In the absence of CHCA + NaOH, we have the rate of decay of PHEN⁺ as

$$-\frac{d[\text{PHEN}^+]}{dt}\Big|_{\{(1)-(2)\}} = k_{(1)}[\text{PHEN}^+][\text{O}_2^-] + k_{(2)}[\text{PHEN}^+][\text{DCB}^-] \dots\dots\dots(i)$$

In the presence of CHCA + NaOH, we have

$$-\frac{d[\text{PHEN}^+]}{dt}\Big|_{\{(1)-(4)\}} = k_{(1)}[\text{PHEN}^+][\text{O}_2^-] + k_{(2)}[\text{PHEN}^+][\text{DCB}^-] + k_{(3)}[\text{PHEN}^+][\text{OH}^-] + k_{(4)}[\text{PHEN}^+][\text{RCOO}^-] \dots\dots(ii)$$

Under the conditions of the photocatalytic decarboxylation reaction,

$$[\text{RCOO}^-]; [\text{OH}^-] \gg [\text{PHEN}^+]; [\text{DCB}^-]; [\text{O}_2^-]$$

because the latter set of species is formed due to electron transfer after photoexcitation of PHEN. We can assume that [RCOO⁻] and [OH⁻] do not change appreciably during the course of the reaction.

Eqn. (ii) therefore simplifies to

$$-\left.\frac{d[PHEN^+]}{dt}\right|_{\{(1)-(4)\}} = k_{(1)}[PHEN^+][O_2^-] + k_{(2)}[PHEN^+][DCB^-] + \{k'_{(3)} + k'_{(4)}\}[PHEN^+] \dots (iii)$$

Here, $k'_{(3)} = k_{(3)}[OH^-]$ and $k'_{(4)} = k_{(4)}[RCOO^-]$ are the pseudo, first-order reaction coefficients for reactions (3) and (4), respectively.

The difference between the rates of decay of $PHEN^+$ with and without the presence of CHCA + NaOH, [(iii) – (i)], is

$$-\left.\frac{d[PHEN^+]}{dt}\right|_{\{(1)-(4)\}} + \left.\frac{d[PHEN^+]}{dt}\right|_{\{(1)-(2)\}} \simeq \{k'_{(3)} + k'_{(4)}\}[PHEN^+] \dots (iv)$$

Experimentally, we find this difference to be approximately 100 ns (e.g., the rates are 230 ± 70 ns and 340 ± 70 ns, respectively, with and without CHCA + NaOH). We use the faster decay rate in the kinetic model because the fate of the decarboxylation step is determined by the overall decay of $PHEN^+$. The lifetime of the radical ($\tau_2 = 1/k_2$) is solely determined by the risetime of CO_2^- the mean and standard deviation over nine such measurements are used to evaluate the lifetime of the carboxyl radical (500 ns) and the error bar, respectively (± 120 ns).

The non-reactive decay channels of $PHEN^+$ somewhat limit the success of the decarboxylation step in our experiment. Accordingly, the estimated appearance time of the radical is determined by the overall decay of $PHEN^+$ induced by both slower reactive and faster non-reactive channels. The time-window over which the radical is populated is likely to be extended to microseconds if O_2 is carefully removed and / or back electron transfer from DCB^- is inhibited by adding water for the efficient solvation and separation of the ion pair. Notwithstanding a precise knowledge of the O_2 concentration in the reaction medium, the kinetic model presented in Figure 4 is robust and can be formally applied to estimate the lifetimes of the reactive intermediates under different reaction conditions.”

Comment: The following two technical issues should also be addressed:

(1) Why is the rise and decay of the $PHEN^+$ reported with an integrated absorbance that includes $PHEN^*$ (1550-1580 cm^{-1})? It seems a better and cleaner peak for $PHEN^+$ is 1520 cm^{-1} .

Response: The peaks due to $PHEN^*$ have very weak absorption and do not dominate the kinetics beyond 100 ps. We observe similar decay kinetics of $PHEN^+$ (in the range of 300 to 350 ns) irrespective of which peak(s) are taken into account for the integrated absorbance. Further, as the bimolecular electron transfer events occur on much longer timescales (compared to $PHEN^*$ decay), it does not affect the extraction of the lifetimes of reactive intermediates for which only the risetime of CO_2^- matters.

We now explicitly state this in line 103, “Note that the overlapping peaks due to $PHEN^*$ have weaker absorption and do not dominate the kinetics beyond 100 ps.”

(2) It is not clear how the kinetic model for the radical lifetime includes the O_2 quenching or decay pathway in the absence of CHCA. I believe it should be included if that is a competing pathway. Since this is an additional limitation to directly measuring the lifetime of the radical, it should be shown/discussed in the main manuscript.

Response: This is now explained with the addition of a note in the SI, as explained above. We also have new measurements on the lifetime of the carboxyl radical.

Reviewer #2 (Remarks to the Author):

This manuscript presents an interesting time-resolved spectroscopy investigation employing a 100 kHz mid-infrared probe in a transient absorption spectroscopy experiment to directly observe the decarboxylation of cyclohexanecarboxylic acid (CHCA) in acetonitrile-d₃ from the picosecond to the millisecond timescales by monitoring a photooxidant pair (phenanthrene, PHEN and 1,4-dicyanobenzene, DCB). The selective excitation of PHEN in the reaction mixture at 256 nm allows a diffusion-limited photoinduced electron transfer to DCB and an observed time offset in the rise of the CO₂ byproduct with respect to the decay of the oxidizing arene cation directly measures the lifetime (520 ± 120 ns) of a reactive carboxyl radical in this photocatalytic decarboxylation reaction. This ability to perform accurate spectroscopic lifetime measurements of the radicals produced in situ by an activated C-C bond fission can be utilized to improve photocatalytic selectivity and turnover in related reactions. The experiments and their data analysis and interpretation all have appeared to have been carefully performed and considered. I recommend publication after minor revision to address the comments/suggestions below:

Comment: 1. It would be nice to include some related computational work to support the assignments of the species observed in the time-resolved experiments and to also explore the reaction pathways and barriers proposed for the interpretation of the experimental data. I would encourage the authors to do this to make the research both stronger and more interesting to a broader readership so as to make it more suitable for Nature Communications.

Response: The computational work requested by the reviewer was included in the Supporting Information of our original submission, and cross-referenced in the manuscript. The assignments of the IR bands for the PHEN cation, DCB anion, and anthracene cation are based on quantum chemical calculations at the ω B97xD/6-311++G** level of theory. These results have been included in Figures S5 and referenced in the main paper. The computed energy barrier for the decarboxylation reaction at the same level of theory is provided in Figure S8 and referenced in the main paper.

In view of the referee's comment, we have now included a table of contents and list of figures at the beginning of the Supporting Information to improve the clarity of the presentation. At the end of the computational methodology section, we also write, "*The computational results are shown in Figures S5 and S8.*"

Reviewer #3 (Remarks to the Author):

Time-resolved infrared spectroscopy is applied to study the photochemical decarboxylation of a particular carboxylic acid (CHCA) in acetonitrile using a photooxidant organic donor-acceptor pair, phenanthrene (PHEN) and 1,4-dicyanobenzene (DCB). This reaction is studied over about 12 orders of magnitude in time, from picoseconds to milliseconds.

Comment: The general selling point is that by understanding the reaction mechanisms may be of use in improving molecular design of supramolecular structures. The procedure is argued to occur under milder reaction conditions, using a clean phototrigger, and the use of toxic metal catalysts is avoided. This may be so, however, after the introduction and a lengthy detailed description of the spectroscopic results, this general context is not being mentioned anymore. What does one learn from the results obtained for future applications?

Response: We had briefly presented the potential applicability of our probing methodology in the conclusion, "The results afford a combined kinetic and mechanistic understanding of a four-step photoredox chemical reaction over picosecond to millisecond dynamic time range in a single time-resolved experiment. Our work demonstrates the value of transient absorption with high repetition-rate spectroscopic probes to resolve multi-step photocatalytic reaction mechanisms. A precise knowledge of the mechanistic pathway and lifetimes of the transient radicals or intermediates, afforded by such studies, will lead to further innovation in the design of photoredox-catalysed, self- and cross-coupling reactions."

However, in view of the referee's comments, we have now further elaborated on the general context of our method in new lines 281-286 appended to the preceding paragraph as, "*The methodology outlined here as a proof-of-principle experiment can be readily extended to other photoredox chemical reactions that involve multiple steps or intermediates, such as organic polymerization reactions triggered by the photoinduced generation of free radicals. Our experimental scheme additionally reveals that organic synthetic methods, which traditionally use high power, broadband arc lamps for high product yields may further benefit from low power, narrowband photoexcitation for implementing selectivity in photoredox reactions.*"

Despite the extensive description of the experimental results and the interpretations of those, I sense that many issues remain unmentioned unanswered:

Comment: 1-what are the driving forces for the two charge transfer reactions

PHEN* + DCB - PHEN.+ + DCB.-

PHEN+ + CHCA- - PHEN + CHCA.

would the observed rates makes sense when comparing previously reported results of photoinduced electron transfer reactions of organic donor-acceptor pairs in acetonitrile, as extensively documented by the Vauthey group?

Response: The corresponding redox potential values provide the driving forces for the charge transfer reactions. Ground-state redox potential values (with respect to saturated calomel electrode) of PHEN and DCB are 1.5 V and -1.46 V, respectively. The triplet state energy of PHEN* for 1-electron oxidation is ~2.6 V. In photoexcited PHEN, the electron transfer is barrierless and we find that the diffusion-limited photoinduced electron transfer rates between DCB and PHEN* are indeed consistent with results of the Vauthey group on the pyrene-DCB system. This work has been cited (Reference 16).

Ground-state redox potential values (with respect to saturated calomel electrode) of PHEN and a similar carboxylate (hexanoate) anion are 1.5 V and 1.16 V, respectively. The favourable ΔG values for the decarboxylation reaction are already stated in lines 190-192, "*the secondary electron transfer from a carboxylate ion (eg. hexanoate ion) to PHEN⁺ is a spontaneous process ($\Delta G = -0.34$ V in ACN) but energetically uphill for the anthracene cation ($\Delta G = +0.07$ V in ACN).*" This single electron transfer step for the decarboxylation reaction is spectroscopically observed for the first time in the present study.

Comment: 2- Fig. 1 suggests unit quantum yields for the sequential charge transfer steps. I miss a dedicated discussion of and a quantitative assessment of quantum yields the two charge transfer reactions. The authors need to come with convincing arguments why a backelectron transfer reaction $\text{PHEN}^+ + \text{DCB}^- \rightarrow \text{PHEN} + \text{DCB}$ does not form a serious contender cutting short the decarboxylation process.

Response: IR oscillator strengths for a particular mode can change appreciably upon the addition or removal of an electron. Quantum chemical calculations, although useful in qualitatively predicting the overall trends in the change of absorption frequency or oscillator strength, cannot be used reliably for quantitative estimations of the quantum yield through peak amplitudes, unfortunately. Therefore, the peak amplitudes from the spectroscopy cannot be used at present for a meaningful evaluation of the quantum yields of each electron transfer step.

In principle, the quantum yield of the charge transfer steps is tractable through the recovery of the ground-state depletion of PHEN^* , in the absence of side-reactions such as quenching of PHEN^+ due to O_2^- . However, in Fig. 2(a) we see that the ground-state depletions of PHEN at 1462 and 1506 cm^{-1} recover completely within ~ 100 ns due to O_2^- quenching / charge recombination with DCB^- and this further precludes a meaningful estimation of the quantum yields for both $\text{PHEN}^+ + \text{DCB}^- \rightarrow \text{PHEN} + \text{DCB}$ and $\text{PHEN}^+ + \text{CHCA}^- \rightarrow \text{PHEN} + \bullet\text{CHCA}$. Nonetheless, the experimental results point to two clear conclusions: (lines 201-203), “...a SET between the carboxylate ion and the PHEN^+ cation (step 3) must necessarily occur within the solution-phase lifetime of the PHEN^+ cation for the reaction to proceed through to the final decarboxylation step (step 4)” via control experiments using anthracene and in lines 190-192, “the secondary electron transfer from a carboxylate ion (eg. hexanoate ion) to PHEN^+ is a spontaneous process ($\Delta G = -0.34$ V in ACN) but energetically uphill for the anthracene cation ($\Delta G = +0.07$ V in ACN).”

Back electron transfer from DCB^- to PHEN^+ is indeed expected to be detrimental for the decarboxylation reaction. However, the non-identical decay timescales for PHEN^+ (300 ns) and DCB^- (40 ns) in Figure 2f show that back electron transfer is not the only channel under these reaction conditions that inhibits the decarboxylation step, rather quenching due to trace amounts of oxygen also plays a role. We have now introduced a new note in the SI to show how additional quenching / back-electron transfer reactions affect the estimation of the timescales over which the radical occurs. Note that it does not affect the extraction of the radical lifetime which is evaluated from the experimentally observed risetime of CO_2 . We also propose that effective solvation of the ions, as studied in the organic literature by addition of trace amounts of water, can prevent the back electron transfer and enhance the decarboxylation yield (reference 18 in the SI). Specifically, we write, “The non-reactive decay channels of PHEN^+ somewhat limit the success of the decarboxylation step in our experiment. Accordingly, the estimated appearance time of the radical is determined by the overall decay of PHEN^+ induced by both slower reactive and faster non-reactive channels. The time-window over which the radical is populated is likely to be extended to microseconds if O_2 is carefully removed and / or back electron transfer from DCB^- is inhibited by adding water for the efficient solvation and separation of the ion pair. Notwithstanding a precise knowledge of the O_2 concentration in the reaction medium, the kinetic model presented in Figure 4 is robust and can be formally applied to estimate the lifetimes of the reactive intermediates under different reaction conditions.”

Comment: 3- CHCA is prepared in the anionic form CHCA^- using equimolar amounts of NaOH. May I assume this is anhydrous NaOH? Even then, How much exactly of CHCA is in the anionic form? And because of that one expects to have the same amount of H_2O in solution as well. What is the role of H_2O in the decarboxylation process?.. Is it possible to investigate possible H/D isotope effects (by using NaOD/ deuterated CHCA) to explore the role of water hydrogen bonded to a possible $\text{PHEN}^+ / \text{CHCA}^-$ reactive complex.

Response: We carry out our transient spectroscopy studies under conditions which mimic the optimized conditions used for synthetic applications of the photoredox catalysed reaction to ensure our mechanistic and kinetic deductions are most applicable to synthetic chemists. Hence we use

anhydrous NaOH and we expect to have trace quantities of water in our reaction due to the deprotonation of CHCA in an alkaline medium as the reviewer notes. The pK_a of CHCA in acetonitrile is 7.6, i.e. $K_a = 2.5 \times 10^{-8}$. In an equimolar mixture of NaOH and CHCA, we expect the acid/base equilibrium to lie towards near-complete deprotonation of the acid. If the presence of trace water influences the observed kinetics, then this influence will be the same in our transient absorption studies as in the applications of this chemistry in synthesis.

We can speculate about how water might play a role in the mechanism, but we do not have direct evidence to back up this speculation so we prefer not to discuss it in the main text. The role of water in stabilizing the $\text{PHEN}^+/\text{CHCA}^-$ reactive complex is an interesting aspect of this reaction. However, this is not a hindrance to the decarboxylation reaction because water has earlier been shown to prevent back electron transfer between PHEN^+ and DCB^- by effective solvation and separation of the charged species. This effect is reflected in higher decarboxylation yields in the presence of water (example, using 9:1 $\text{CH}_3\text{CN}:\text{H}_2\text{O}$ mixture as solvent in reference 18 in the SI). We have now added a statement in the new note in the SI to point out the potential role of water in influencing the reaction pathway, *“The time-window over which the radical is populated is likely to be extended to microseconds if O_2 is carefully removed and / or back electron transfer from DCB^- is inhibited by adding water for the efficient solvation and separation of the ion pair. Notwithstanding a precise knowledge of the O_2 concentration in the reaction medium, the kinetic model presented in Figure 4 is robust and can be formally applied to estimate the lifetimes of the reactive intermediates under different reaction conditions.”*

Furthermore, water molecules can, in principle, engage in a hydrogen bond to the reactive $\text{RCOO}\cdot$ intermediate which might modify the barrier for the unimolecular dissociation. We have not computed the effects of water complexation on the unimolecular barrier height, or attempted to quantify the effects further because we do not know the proportion of RCO_2 radicals in solution which are complexed or uncomplexed to water.

We do not expect isotope effects will be observed in any of the kinetic steps studied because the motions of the H atoms in the water molecules are not part of the reaction coordinate (which corresponds primarily to C-C bond fission, with some change in O-C-O bond angle).

We agree with the reviewer that the microsolvation effects of water are an interesting topic for future study and we plan to undertake more experiments in this direction to explore the role of water and other solvents with high dielectric constants in influencing the barrier height along the RCO_2 decomposition reaction coordinate.

List of Other Changes Made:

Line 13: inserted *“Direct spectroscopic observation of the carboxyl radical confirms its formation as a reaction intermediate.”*

Line 38: changed to *“However, with the current research focus on increasing the reaction substrate scope and selectivity...”*

Line 69: inserted *“carboxyl stretch (~1700 cm⁻¹)”*

Line 75: inserted *“The kinetic analysis is also supported by a direct spectroscopic observation of the carboxyl radical intermediate. These results...”*

Line 199: inserted *“We carried out a control experiment with DCB, photoexcited at 280 nm (in the absence of PHEN), in ACN-d₃ containing CHCA / NaOH (24 mM). The experiment showed the formation of DCB* due to photoexcitation but no evidence for the growth of DCB[•] due to a SET from carboxylate anion to DCB*.”*

Line 222: inserted *“total (reactive + non-reactive) decay of PHEN⁺”*

Line 270: inserted *“or DCB* in the absence of PHEN”*

Line 303: inserted *“Data Availability:*

Data are available at the University of Bristol data repository, data.bris, at <https://doi.org/10.5523/bris.10.5523/bris.1q3gfn8eb06hh22sjzql2cu1qg>. All reported data are available from the corresponding authors upon reasonable request”

Line 414: inserted *“and ULTRA facilities”*

REVIEWERS' COMMENTS:

Reviewer #1 (Remarks to the Author):

The authors have done a great job addressing the concerns, and have indeed performed new experiments to do so adequately. Nonetheless, the following two concerns that stem from the revisions should be addressed prior to publication:

(1) It is not clear why the evidence of the carboxyl is buried in the supplementary information. If it is the direct evidence of the intermediate and inferred time-scale, should it not be shown within the main manuscript, even if seen at a higher fluence?

(2) Given the claim in the manuscript of following a catalytic cycle over many orders of magnitude time, and directly observing the time-evolution of intermediates, it would make sense for the following previous work in this direction to be referenced:

On the oxyl radical within water oxidation, over many decades in time:

D.M. Herlihy, M.M. Waegle, X. Chen, C.D. Pemmaraju, D. Prendergast and T. Cuk, "Detecting the Oxyl Radical of Photocatalytic Water Oxidation by its Sub-Surface Vibration" *Nature Chemistry* 2016, 8, 549.

X. Chen, S. Choing, D. Aschaffenburg, C.D. Pemmaraju, D. Prendergast and T. Cuk, "The Formation Time of Ti-O• and Ti-O•-Ti Radicals at the n-SrTiO₃/Aqueous Interface during Photo-catalytic Water Oxidation" *J. Am. Chem. Soc.*, 2017, 139, 1830

X. Chen, D. Aschaffenburg, and T. Cuk, "Selecting between two transition states by which water oxidation intermediates on an oxide surface decay", *Nat. Catalysis*, 2019, 2, 820

On the transition metal oxo within water oxidation, at millisecond and steady state time scales:

Zhang, M., de Respinis, M. & Frei, H. Time-resolved observations of water oxidation intermediates on a cobalt oxide nanoparticle catalyst. *Nat. Chem.* 6, 362-367, (2014).

Zandi, O. & Hamann, T. W. Determination of photoelectrochemical water oxidation intermediates on haematite electrode surfaces using operando infrared spectroscopy. *Nat. Chem.* 8, 778-783, (2016).

All of the above are on surfaces and on a different reaction, but the principle is there. I would also suggest referencing earlier work by Charles Harris on time-resolved IR of photo-excited organic molecules in solution, undergoing reaction steps. I leave it to the authors how best to do this, given their results and the earlier work by Harris and co-workers.

Reviewer #2 (Remarks to the Author):

The authors have satisfactorily addressed the points raised in my previous review and I think the present form of the manuscript can be published.

Reviewer #3 (Remarks to the Author):

In my opinion the authors have responded more than averaged to my in depth comments on data analysis, some of which were indeed of serious level. My questions on quantum yields in particular have been met with the right amount of arguments being available. I see many interesting general questions arising out of this work, that likely will lead to further studies on condensed phase electron transfer reactions.

The revised manuscript meets now the standard criteria for Nature Commun. and I strongly recommend publication!

Another reason why this work deserves publication is that the equipment used, 100 kHz Yb:KGW amplifier pumping a UV/mid-IR set-up with sensitivities of measuring absorbance changes as small as 0.02 mOD. This newest state-of-the-art achieved by the RAL team deserves further exposure by results achieved with it, as evidenced with this manuscript. Both the RAL and Bristol teams should not be too hesitant to further advertise this through informal and formal channels. (I sincerely hope press releases will get noticed in the current Uncertain Times).

Reviewers' comments:

Reviewer #1 (Remarks to the Author):

The authors have done a great job addressing the concerns, and have indeed performed new experiments to do so adequately. Nonetheless, the following two concerns that stem from the revisions should be addressed prior to publication:

Comment: (1) It is not clear why the evidence of the carboxyl is buried in the supplementary information. If it is the direct evidence of the intermediate and inferred time-scale, should it not be shown within the main manuscript, even if seen at a higher fluence?

Response: The choice of placing the additional experiment in the Supporting Information is based on two considerations:

(1) So as not to confuse the readers about this dataset being the same experiment as reported in Figure 3 of the main paper. This experiment was conducted at higher pump fluence, using a different instrument even, and we feel this distinction ought to be made clear without the readers having to figure the subtleties themselves. The different extent of oxygen contamination in the two sets of measurements results in expected discrepancies in the build-up time for the radical, as already explained in the Supplementary Materials. We therefore prefer to show it as a supporting evidence to validate our kinetic model, and limit the main text to mention of the position of the IR band of the carboxyl radical and its general kinetic behaviour.

(2) To reflect the predictive power of a kinetic model in estimating the lifetimes of reactive intermediates in complex multistep reactions, even if not directly observed, as we already argued in lines 246-262 in the last revision. While obtaining a direct evidence for a reaction intermediate in any experiment is always definitive, technical difficulties that preclude direct observation may be circumvented by the judicious use of kinetic models that make use of empirical timescales for reactant (decay) and product (rise) as the only parameters. This extends the general applicability of the probing method to other photocatalytic decarboxylation reactions.

Comment: (2) Given the claim in the manuscript of following a catalytic cycle over many orders of magnitude time, and directly observing the time-evolution of intermediates, it would make sense for the following previous work in this direction to be referenced:

On the oxyl radical within water oxidation, over many decades in time:

D.M. Herlihy, M.M. Waagele, X. Chen, C.D. Pemmaraju, D. Prendergast and T. Cuk, "Detecting the Oxyl Radical of Photocatalytic Water Oxidation by its Sub-Surface Vibration" *Nature Chemistry* 2016, 8, 549.

X. Chen, S. Choing, D. Aschaffenburg, C.D. Pemmaraju, D. Prendergast and T. Cuk, "The Formation Time of Ti-O• and Ti-O•-Ti Radicals at the n-SrTiO₃/Aqueous Interface during Photo-catalytic Water Oxidation" *J. Am. Chem. Soc.*, 2017, 139, 1830

X. Chen, D. Aschaffenburg, and T. Cuk, "Selecting between two transition states by which water oxidation intermediates on an oxide surface decay", *Nat. Catalysis*, 2019, 2, 820

On the transition metal oxo within water oxidation, at millisecond and steady state time scales:

Zhang, M., de Respinis, M. & Frei, H. Time-resolved observations of water oxidation intermediates on a cobalt oxide nanoparticle catalyst. *Nat. Chem.* 6, 362-367, (2014).

Zandi, O. & Hamann, T. W. Determination of photoelectrochemical water oxidation intermediates on haematite electrode surfaces using operando infrared spectroscopy. *Nat. Chem.* 8, 778-783, (2016).

All of the above are on surfaces and on a different reaction, but the principle is there. I would also suggest referencing earlier work by Charles Harris on time-resolved IR of photo-excited organic molecules in solution, undergoing reaction steps. I leave it to the authors how best to do this, given their results and the earlier work by Harris and co-workers.

Response: Some of these representative publications are now cited. Specifically, these are new references 14 and 15 in the main paper.

We thank the referee for their careful reviewing of the manuscript.

Reviewer #2 (Remarks to the Author):

The authors have satisfactorily addressed the points raised in my previous review and I think the present form of the manuscript can be published.

Response: We thank the referee for their comments on the manuscript.

Reviewer #3 (Remarks to the Author):

In my opinion the authors have responded more than averaged to my in depth comments on data analysis, some of which were indeed of serious level. My questions on quantum yields in particular have been met with the right amount of arguments being available. I see many interesting general questions arising out of this work, that likely will lead to further studies on condensed phase electron transfer reactions. The revised manuscript meets now the standard criteria for Nature Commun. and I strongly recommend publication!

Another reason why this work deserves publication is that the equipment used, 100 kHz Yb:KGW amplifier pumping a UV/mid-IR set-up with sensitivities of measuring absorbance changes as small as 0.02 mOD. This newest state-of-the-art achieved by the RAL team deserves further exposure by results achieved with it, as evidenced with this manuscript. Both the RAL and Bristol teams should not be too hesitant to further advertise this through informal and formal channels. (I sincerely hope press releases will get noticed in the current Uncertain Times).

Response: We thank the referee for their helpful comments and suggestions.